# *Thymus vulgaris* Essential Oil in Beta-Cyclodextrin for Solid-State Pharmaceutical Applications

**DOI:** 10.3390/pharmaceutics15030914

**Published:** 2023-03-11

**Authors:** Aldo Arrais, Elisa Bona, Valeria Todeschini, Alice Caramaschi, Nadia Massa, Maddalena Roncoli, Alessia Minervi, Elena Perin, Valentina Gianotti

**Affiliations:** 1Dipartimento per lo Sviluppo Sostenibile e la Transizione Ecologica, Università del Piemonte Orientale, 13100 Vercelli, Italy; 2Dipartimento di Scienze e Innovazione Tecnologica, Università del Piemonte Orientale, 15121 Alessandria, Italy

**Keywords:** red thyme, essential oil, antibacterial assay, antifungal assay, *Candida albicans*, *C. glabrata*, *Staphylococcus aureus*, *Pseudomonas aeruginosa*, *Klebsiella pneumoniae*

## Abstract

Antimicrobial resistance related to the misuse of antibiotics is a well-known current topic. Their excessive use in several fields has led to enormous selective pressure on pathogenic and commensal bacteria, driving the evolution of antimicrobial resistance genes with severe impacts on human health. Among all the possible strategies, a viable one could be the development of medical features that employ essential oils (EOs), complex natural mixtures extracted from different plant organs, rich in organic compounds showing, among others, antiseptic properties. In this work, green extracted essential oil of *Thymus vulgaris* was included in cyclic oligosaccharides cyclodextrins (CD) and prepared in the form of tablets. This essential oil has been shown to have a strong transversal efficacy both as an antifungal and as an antibacterial agent. Its inclusion allows its effective use because an extension of the exposure time to the active compounds is obtained and, therefore, a more marked efficacy, especially against biofilm-producing microorganisms such as *P. aeruginosa* and *S. aureus*, was registered. The efficacy of the tablet against candidiasis opens their possible use as a chewable tablet against oral candidiasis and as a vaginal tablet against vaginal candidiasis. Moreover, the registered wide efficacy is even more positive since the proposed approach can be defined as effective, safe, and green. In fact, the natural mixture of the essential oil is produced by the steam current method; therefore, the manufacturer employs substances that are not harmful, with very low production and management costs.

## 1. Introduction

Antimicrobial resistance linked to the misuse of antibiotics in the modern era is a well-known topic. Their excessive use in several fields, such as intensive farming and medicine, has led to enormous selective pressures on pathogenic and commensal bacteria, driving the evolution of antimicrobial resistance genes [1,2,3] with severe impacts on human health. In fact, bacteria are able to overcome antibiotic effects thanks to the expression of antibiotic-resistance genes employing an efflux pump or enzymatic deactivation of antibiotic molecules and chemical modification of their cellular targets [4]. The direct consequence is that the therapeutical efficacy of antibiotic drugs [3] has decreased over time, and it is set to get even worse. It is assumed that by 2050, 10 million victims per year will occur due to drug-resistant pathogens.

In this context, a list of six human pathogenic bacteria, typically associated with nosocomial infections, was identified with the acronym ESKAPE: *Enterococcus faecium*, *Staphylococcus aureus*, *Klebsiella pneumoniae*, *Acinetobacter baumannii*, *Pseudomonas aeruginosa*, and *Enterobacter species* [5,6]. These bacteria are identified as ESKAPE since they are able to “escape” the antibiotic activity. For example, according to the document of the European Centre for Disease Prevention and Control (ECDC) (https://www.ecdc.europa.eu/sites/default/files/documents/surveillance-antimicrobial-resistance-Europe-2019.pdf (accessed on 9 November 2021)), in 2019, more than 33% of the *K. pneumoniae* isolates showed resistance to at least one of the antibacterial groups under surveillance and often were characterized by a combined resistance to different antimicrobial groups. In several European countries, *K. pneumonia* and *P. aeruginosa* showed percentages of carbapenem resistance above 10%, reaching over 35% in Italy. Moreover, methicillin-resistant *S. aureus* (i.e., MRSA) is also considered an important pathogen in European countries, showing combined resistance to other antimicrobial groups. It is necessary to develop new drugs for these species, as proposed by the World Health Organization (WHO) [7].

Moreover, yeasts, such as *Candida* spp., cause infections localized in the mouth, skin, and vagina that can also become systemic. These infections lead to more than 3.6 million healthcare visits each year in the U.S., and the estimated direct medical costs amount to USD 3 billion [8]. Most candidiasis are caused by *Candida albicans*, a yeast with lower antibiotic resistance features; other species, such as *C. glabrata* and *C. auris,* are frequently more resistant and deadly [9]. Azoles and polyenes are the two groups of drugs specially used against yeasts, but now these molecules are not always successful due to the resistance developed by the *Candida* sp. strains.

It, therefore, becomes crucial to implement alternative strategies both in terms of scientific biomedical research and the socio-political and economical perspectives. Among all the possible strategies, a viable one could be the development of medical features that employ natural products [10] since they overcome many limitations of synthetic pharmaceutics (i.e., the environmental impact of chemical syntheses and the high economic costs of processes) [10,11]. If properly investigated, they could be competitive regarding clinical performance.

Among natural products, essential oils (EOs) extracted from different plant organs, such as leaves, flowers, fruits, seeds, roots, buds, stems, and wood [12] represent a good resource as they are complex natural mixtures of organic compounds showing, among others, antiseptic properties. In fact, their inhibitory activities against fungal and bacterial pathogens are reported in the literature [13,14,15,16,17], especially for EOs with high concentrations of phenols. While the properties of EOs extracted from medicinal plants (such as oregano and winter savory) are well documented in the literature [15,17], those of thyme have been little investigated. On the contrary, the antimicrobial effect of red thyme has been scarcely reported in the literature [18,19,20]. The natural EOs’ antiseptic performances against bacterial and fungal strains mimic those of antibiotics applied in clinical routine [15,17,21]. Unfortunately, despite these effective bioactive properties, their oily nature, especially high volatility, and low aqueous solubility pose serious limitations both in their employment as medical devices and in their potential therapeutic application [22].

The use of inert solid carriers such as cyclic oligosaccharides cyclodextrins (CD), able to incorporate hydrophobic substances into their cavity, can be a valid approach to overcome these limitations [22,23]. The application of CD and their derivatives in pharmaceutical preparations has a positive impact since they are non-toxic elements with high biocompatibility and solubility in water [24,25]. A successful application of such an approach was the inclusion of oregano and winter savory EOs of in beta-cyclodextrins (b-CD), the most used natural cyclodextrins, obtaining a transfer of the EO liquid components in a solid form [21].

This work has two main purposes. First, to go deeper into the chemical composition and the biological activity against bacteria and yeasts of red thyme (*Thymus vulgaris*) EO and second, to expand the study of the inclusion approach in solid carriers through an extensive characterization of the obtained product in order to broaden the applicability of host–guest inclusion complexes. In particular, red thyme EO was employed as an antimicrobial agent and beta-cyclodextrins (b-CD) as a solid carrier.

## 2. Materials and Methods

### 2.1. Thymus vulgaris Oil Embedding in b-CD Procedure 

The *T. vulgaris* oil under investigation was purchased by Flora s.r.l. (Pisa, Italy). 

The EO-b-CD complex synthesis was obtained using an ultrasound (Argolab DU-32, Arezzo, Italy) with direct processing of 2 h at room temperature at the higher 5 power level. The weight ratio was 500 mg of EO mixed with 750 mg of b-CD. After the treatment, the solid-state materials were washed with 2 mL of deionized water (to remove unreacted cyclodextrins) and dried under a ventilated hood for 48 h. 

The obtained solid-state composite powder was obtained by embedding, in an agate mortar, 20 mg of the EOs-b-CD powder in polyvinylpyrrolidone (PVP) (45 mg). Then, a 1 cm diameter round tablet (5 mm thickness) was prepared by pressing for 1 min at 10 Ton/cm^2^ the composite powder obtaining A proto-pharmaceutical formulation.

### 2.2. FT-IR Analyses 

Solid-state Fourier-Transformed (FT) InfraRed (IR) spectra were collected in dry KBr 1-cm pressed (10 tons per square centimeter) discs with a Thermo-Fisher Scientific Nicolet iS50 spectrophotometer (1–3 mg of embedded solid-state samples; for the liquid EO, the surface of a blank KBr disc was impregnated by 1 drop of the essential oil), at 2 cm^−^^1^ spectral resolution of the collected interferogram (100 averaged scansions). This parameter was observed to be appropriate for detecting unambiguously the host–guest complexation peak shifts correlated to the modified intermolecular forces, typically occurring in the 1–15 cm^−^^1^ energy intervals [21,26,27].

### 2.3. Thermo-Gravimetrical Analyses (TGA)

Thermo-Gravimetrical profiles of Red Thymus essential oil, b-CD, and reacted EO-b-CD inclusion complexes were obtained with a Setaram labSys Evo instrumentation in alumina crucibles under air (25 mL/min) at 5 °C/min heating ramp, in the 25–700 °C thermal interval. In detail, for the EOs, 56.9 mg were analyzed, while for b-CD and EO-b-CD, complex weighted samples were 38.4 and 44.7 mg, respectively.

### 2.4. Gas Chromatography-Mass Spectroscopy (GC-MS)

The chromatographic characterization was performed using the following:-A Gas Chromatograph Finnigan Trace GC-Ultra;-A mass spectrometer Trace DSQ;-A capillary column Phenomenex ZB-WAX (30 m length, 0.25 mm I.D., 0.25 μm film thickness);-Inlet temperature of 250 °C;-Splitless mode;-He as the carrier gas (1.0 mL/min);-Initial oven temperature of 45 °C and the in ramps reported in Table 1;-Mass spectrometer transfer line temperature of 290 °C;-MS signal acquired in El+ mode;-Ionization energy 70.0 eV −;-Source temperature of 290 °C;-Solvent delay 6.50 min;-Mass spectrometric detection 35–500 m/z (full-scan).

The thyme extracts were dissolved in CH_2_CL_2_ (50.00 mg/1.00 mL) filtered (PTFE membrane, 0.20 μm) and analyzed after a 1:5 dilution in CH_2_CL_2_.

**Table 1 pharmaceutics-15-00914-t001:** Oven temperature program for EO.

	Rate (°C/min)	Temperature (°C)	Hold Time (min)
INITIAL		45.0	2.0
RAMP 1	3.0	100.0	0.1
RAMP 2	5.0	135.0	0.1
RAMP 3	8.0	250.0	12.0

### 2.5. Calibration Procedure

The most intense compounds identified both in EO and in the included samples EO-b-CD were quantified. 

The considered analytes were eucalyptol (Cas n° 470-82-6), linalool (Cas n° 78-70-6), (−)-trans-caryophyllene (Cas n° 87-44-5), thymol (Cas n° 89-83-8), and carvacrol (Cas n° 499-75-2). All the standards were purchased from Sigma Aldrich (Milan, Italy). 

Calibration curves were calculated by the injection of a multi-analyte standard at different concentrations, namely 1.00, 5.00, and 10.00 mg/L, and analyzed using the same method employed for the real sample analyses. 

The inclusion percentages were calculated as the ratio between the amount per gram obtained in the EO characterization and the amount obtained in the EO-b-CD sample. 

### 2.6. Antifungal and Antibacterial Activity Assays

The antifungal and antibacterial activity of *T. vulgaris* essential oil and the tablet was assessed with agar disc diffusion following the methods previously published [14,15,16,17]. 

#### 2.6.1. Antifungal Assay

*Candida albicans* ATCC 14,053 and *C. glabrata* ATCC 15,126 reference strains were employed to assess the antifungal activity. Clotrimazole (10 μg) antifungal effects of EO and EO-b-CD were evaluated as reported in the Clinical and Laboratory Standards Institute Standard M44-A. Mueller–Hinton Agar (VWR chemicals, Milan, Italy) added with 2% Glucose and 0.5 μg/mL Methylene Blue Dye (GMB) was used as medium. Briefly, strain suspensions (10^6^ CFU ml^−1^) were swabbed on the medium surface, filter paper discs (diameter of 6 mm) were placed on the surface and added with 10 μL of the EO. The positive control was clotrimazole (10 μg). Negative controls were 1,4 Dioxane (Sigma-Aldrich, St. Louis, MO, USA; 10 μL) and organic linseed oil (10 μL) discs. Triplicate experiments were performed by incubating plates at 37 °C for 48 h. The sensitivity test for the extract is considered positive if the inhibition halo is higher than that induced by clotrimazole (positive control ≥ 100%). 

#### 2.6.2. Antibacterial Assay

The reference strains *Staphylococcus aureus* NCTC6571, *Pseudomonas aeruginosa* ATCC27853, and *Klebsiella pneumoniae* ATCC13883 were used to test the antibacterial activity of EO. Vancomycin, imipenem, and meropenem effects were evaluated according to the EUCAST Disk Diffusion Method for Antimicrobial Susceptibility v. 7.0. Extract biological activity was assessed with the diffusion method. Suspensions of the different strains (0.5 McFarland) were swabbed on Mueller–Hinton agar medium. Filter paper discs were placed on the medium surface and added with 10 μL of EO suspension. 1,4 Dioxane (10 μL) and organic linseed oil (10 μL) discs were used as negative controls, while vancomycin, meropenem, and imipenem were considered as the positive control. Plates were incubated at 37 °C for 24 h. All experiments were performed in triplicate. The halos were measured in mm using calipers. The extract was evaluated as active when the measured halo was equal to or higher than the positive control (positive control ≥ 100%). 

#### 2.6.3. Statistical Analysis

The disk diffusion results were statistically analyzed using one-way ANOVA followed by Tukey’s HSD multiple comparisons of means using R (v. 3.5.1) [21]. Data are presented as boxplots. Differences were considered significant for *p*-values < 0.05.

## 3. Results and Discussion

The EO-b-CD complex was synthesized using ultrasound with direct processing at ambient temperature employing a molar ratio slightly higher than 1:1 (in favor of the b-CD). Consequently, the molar ratio in the reactive mixture is slightly higher than 1:1 (in favor of the b-CD), potentially also allowing the formation of b-CD:EO 2:1 complexes involving the higher terpenes, however, minor constituents. Moreover, this massive ratio already in the past [21] with other oils has guaranteed an effective and extensive encapsulation of the substantial totality of the oil, considering that in EOs, the bulk of the formulation is monoterpenes and sesquiterpenes.

### 3.1. Essential Oil and Complex Characterisation

All precursors and yielded complexes were exhaustively characterized by Fourier-Transformed Infrared (FT-IR) spectrophotometry, thermogravimetric analysis (TGA), and GC-MS.

FT-IR spectra are reported in Figure 1. The EO pattern is peculiar for a complex mixture of organic terpenoid and volatile compounds, with a plethora of sharp peaks in the diagnostic regions related to aliphatic (and a few aromatic) normal vibrational modes of isoprenoid architectures. Oxygenated moieties can be observed in the strong, broad bands at 3480 and 1630 cm^−^^1^, the latter indented at higher wavenumbers as a consequence of different carbonyl C=O modes (i.e., aldehydes, ketones, and carboxylic groups). The b-CD IR profile is reported, typically consistent with a carbohydrate assembly (large, broad signals mainly due to the oxygenated polar functions). For the EO-b-CD complex, the FT-IR spectrum nearly resembles a combination of the two former profiles, although it is not superimposable. In general, broader bands, accompanied by both moderate frequency shifts (higher than the applied 2 cm^−^^1^ spectral resolution in measurements) and a change in relative intensities, can be observed. These comprehensive vibrational phenomena have been observed in yielded inclusion complexes in cyclodextrins for the modified intermolecular environment after host–guest supramolecular recognition [21,26,27]. 

In detail, the two sharp peaks at 1165 and 1153 cm^−1^ of EO coalesce in one broad at 1155 cm^−^^1^; similarly, for the two EO 1061 and 1053 cm^−^^1^ peaks, in one broader EO-b-CD at 1057 cm^−^^1^; the EO peak 1506 cm^−^^1^ enhances its relative intensity in the b-CD complex; the weak EO peak at 1260 cm^−^^1^ hampers further its relative intensity in the complex; the spectral shoulder at 1214 cm^−^^1^ of the EO band at 1232 cm^−^^1^ is not observed in the broad peak profile of the corresponding b-CD complex signal; the EO peak at 1129 cm^−^^1^ is red-shifted at 1124 cm^−^^1^ in the related complex, whilst the EO signal at 1114 is broader and hampered at 1110 cm^−^^1^ in the complex; 1089 cm^−^^1^ EO peak red-shifts at 1082 in the complex, and its spectral shoulder originally at 1082 cm^−^^1^ is lost in the latter; the strong EO band at 970 cm^−^^1^ (in the diagnostic aromatic out-of-plane bending gamma-C-H modes) is not observed in the related complex; similarly for 920 and 839 cm^−^^1^ EO peaks (minor or difficult inclusion of aromatic hosts inside the lipophilic pocket of b-CD may well account for this spectral effect); finally, the scarce aromatic C-H stretching signals above 3000 cm^−^^1^ cannot be observed in the EO-b-CD-complex; in the bCD TR EO complex (albeit merged with the imposing O-H IR absorption of the host cage). Notably, these latter comprehensive hampering spectral features of the strongly lipophilic groups in obtained complexes can be associated with supramolecular recognition phenomena, in which molecular guests penetrate the cyclodextrin lipophilic pocket by specular insertion of their most lipophilic moieties, surrounding and folding their vibrational activity.

In Figure 2, the FT-IR profile of the EO-b-CD complex embedded in the PVP excipient matrix is reported. A few diagnostic pics of the complexed OE emerge in the large, broad signal of the PVP pattern.

In Figure 3, the diagnostic TGA profiles of the essential oil (A), b-CD (B), and EO-b-CD complex (C) are reported. As expected, the essential oil is highly volatile, and a complete weight loss is observed just above 200 °C. The cyclodextrin pattern after carbohydrate dehydration results is stable up to 300 °C (highlighted by the red ellipse), as previously reported by Abarca et al. [28] and de Santana et al. [29]. The b-CD-EO complex (C) is not a mere superposition of the two parent profiles. Actually, in the 100–300 °C stability zone range of cyclodextrin, a progressive weight loss can be observed due to the more stable inclusion of EO components in the CD lipophylic pocket. 

These results support the previous FT-IR findings in assessing the host–guest actual complexation (highlighted by the red ellipse).

The essential oil (EO) and the EO-b-CD complex were also characterized with gas chromatographic analysis coupled with mass spectrometry (GC-MS).

Table 2 reports the list of molecules identified with GC-MS and the characteristics of each compound in terms of chemical classes with respect to the class to which they belong and the information present in the literature on biocidal capabilities. The volatile profile of *T. vulgaris* is very rich and complex, as more than seventy compounds have been identified. Among these, a very large number have been reported to have biological activity against bacteria and fungi.

### 3.2. Evaluation of the Inclusion

With the aim of completing the characterization of the produced materials, including a quantitative evaluation, the five most intense compounds, namely eucalyptol, linalool, (-)-*trans*-caryophyllene, thymol, and carvacrol identified both in EO and in the included samples EO-b-CD were quantified. Calibration curves were obtained with the injection of a multi-analytes mixture at different concentrations analyzed using the same GC-MS method employed for the real sample analyses. Using the ordinary least square (OLS) method, calibration curves were obtained with an R^2^ always greater than 0.9995. The model was validated by performing an ANOVA procedure. The F value deriving from the regression (FREG) was greater than the tabled F value (Fcrit), pointing out that the response linearity was verified. Moreover, the performed F tests evidence that there is no lack of fit.

In Table 3, the data obtained in the calibration step and the amount of the analyte registered in the EO, in the included sample (EO-bCD), and in the tablet are shown.

The amounts obtained were then compared, taking into account the amount employed in the synthesis to obtain the incorporation percentage. Such values are useful to explain the data of the antifungal and antibacterial assays reported in the following.

### 3.3. Antifungal and Antibacterial Assays

Figure 4 and Figure 5 show the antifungal activity assay results of red thyme oil and tablet. In particular, the essential oil was the most effective in inhibiting the growth of both *C. albicans* and *C. glabrata*. In fact, the essential oil produces an inhibition halo more than double that of the positive control, while the tablet produces an inhibition halo similar to the essential oil as regards *C. albicans*, while for *C. glabrata,* the activity is comparable to that of the positive control (the antifungal drug), even if not statistically significant.

Figure 6 and Figure 7 show the results of the antibacterial test against *S. aureus*, *P. aeruginosa,* and *K. pneumoniae*. In this case, the essential oil showed comparable activity to the positive control for all three bacteria, as well as the tablet. This is a very interesting result, especially considering the efficacy demonstrated by the red thyme EO and the produced tablet against *P. aeruginosa*, which is a very difficult bacteria and normally not sensitive to natural compounds. Moreover, the tablet showed a complete inhibition halo suggesting that the prolonged time of exposure, due to the necessary dissolution time of the tablet, increased the efficacy of the active compounds. In the literature, red thyme essential oil is reported to be applied in food conservation and not for clinical purposes [19,20].

As stated above, the most present compounds in the tablet are monoterpenes and sesquiterpenes, eucalyptol, linalool, thymol, carvacrol, and caryophyllene. Some of them are already reported in the literature to be active against bacteria, others not. For example, thymol, as a critical component of *T. vulgaris* L. essential oil, is reported to combat *P. aeruginosa* by intercalating DNA and inactivating biofilm [67] via the inhibition of quorum sensing [68]. Carvacrol is reported to be active against biofilm formation of *P. aeruginosa* and *S. aureus* [69]; carvacrol reduced the amount of biofilm by up to 91–100% for *P. aeruginosa* and up to 95–100% for *S. aureus* [70]. Thymol is also reported to be effective against *Candida* species [71].

## 4. Conclusions

In conclusion, our work demonstrates the strong transversal efficacy of red thyme both as an antifungal and antibacterial agent.

The inclusion of the thyme oil in the b-CD appears to be very advantageous since the tablets produced allow an extension of the exposure time to the active compounds and, therefore, a more marked efficacy, especially against biofilm-producing microorganisms such as *P. aeruginosa* and *S. aureus*. In addition, pure essential oil often can be irritating if administered directly on the mucous membranes, while if administered in slow release, it can give the effects of reducing the diffusion of the pathogenic agent without unwanted side effects. The efficacy of the tablet against candidiasis is also very interesting. Such efficacy opens the possible use of this pharmaceutical product as a chewable tablet against oral candidiasis and as a vaginal tablet against vaginal candidiasis.

Moreover, the registered wide efficacy is even more positive since obtained using a very simple production method. In fact, the natural mixture of the essential oil is produced by the steam current method; therefore, the manufacturer employs substances that are not harmful and without high production costs, making the proposed approach effective, safe, and green.

## Figures and Tables

**Figure 1 pharmaceutics-15-00914-f001:**
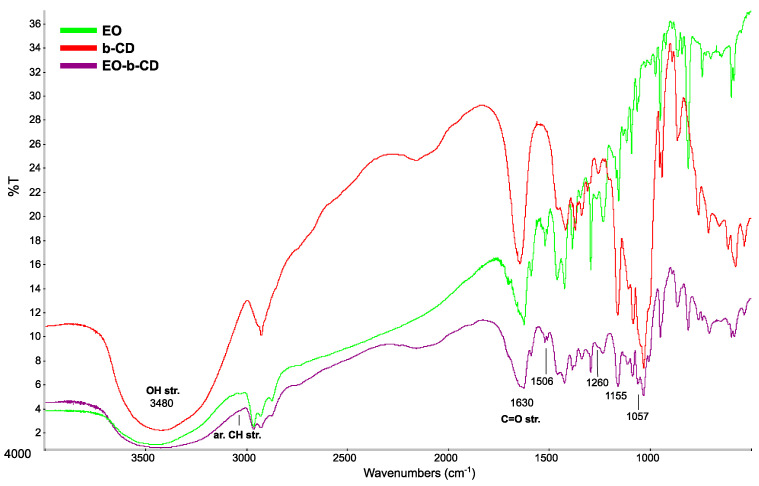
FT-IR spectra of essential oil (green), cyclodextrin (b-CD) (red), and EO-b-CD (purple).

**Figure 2 pharmaceutics-15-00914-f002:**
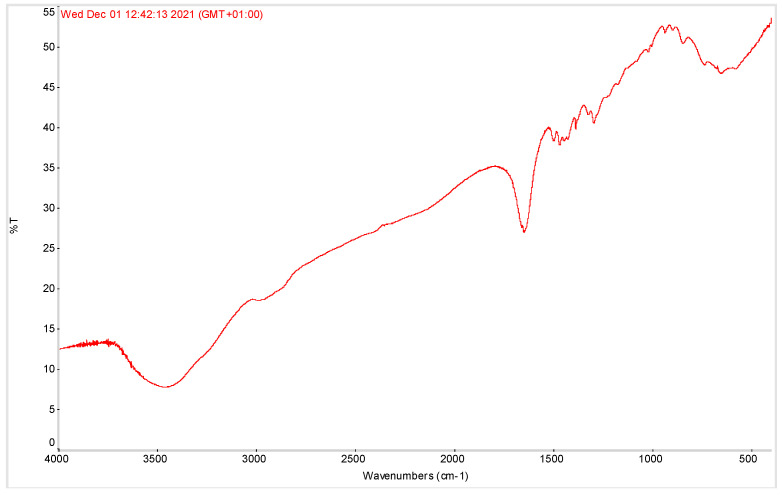
FT-IR profile of the EO-b-CD complex embedded in the PVP excipient matrix.

**Figure 3 pharmaceutics-15-00914-f003:**
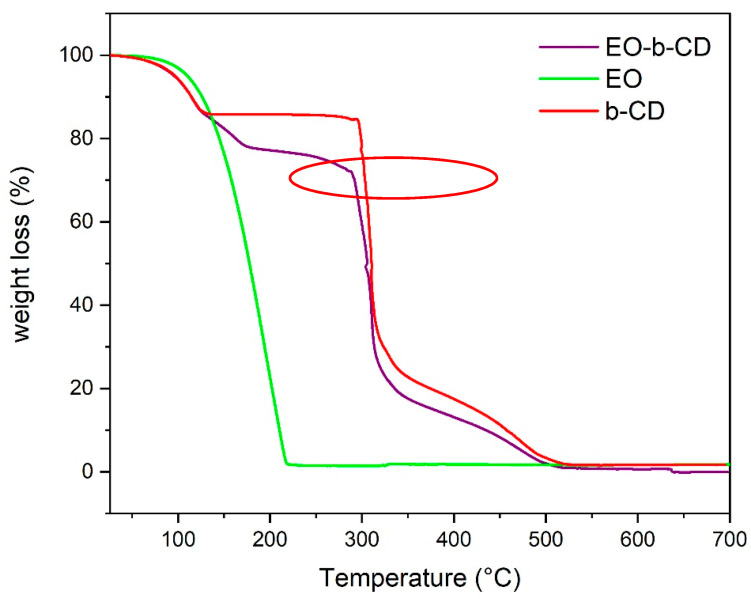
TGA profile of b-CD (red), EO (green), and of the EO-b-CD complex (purple).

**Figure 4 pharmaceutics-15-00914-f004:**
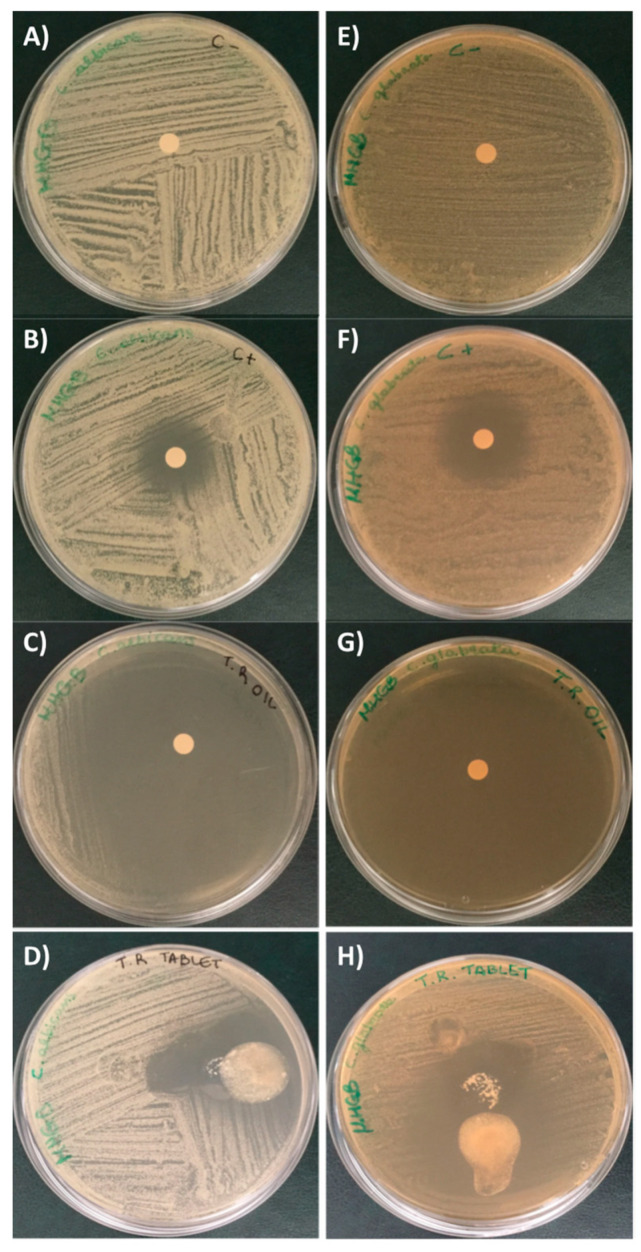
Growth inhibition halo (disk diffusion test) induced by *T. vulgaris* on *C. albicans* and *C. glabrata*. (**A**) *C. albicans* negative control (1,4-dioxane); (**B**) *C. albicans* positive control (clotrimazole); (**C**) *T. vulgaris* essential oil against *C. albicans*; (**D**) Tablet against *C. albicans*; (**E**) *C. glabrata* negative control (1,4-dioxane); (**F**) *C. glabrata* positive control (clotrimazole); (**G**) *T. vulgaris* essential oil against *C. glabrata*; (**H**) Tablet against *C. glabrata*.

**Figure 5 pharmaceutics-15-00914-f005:**
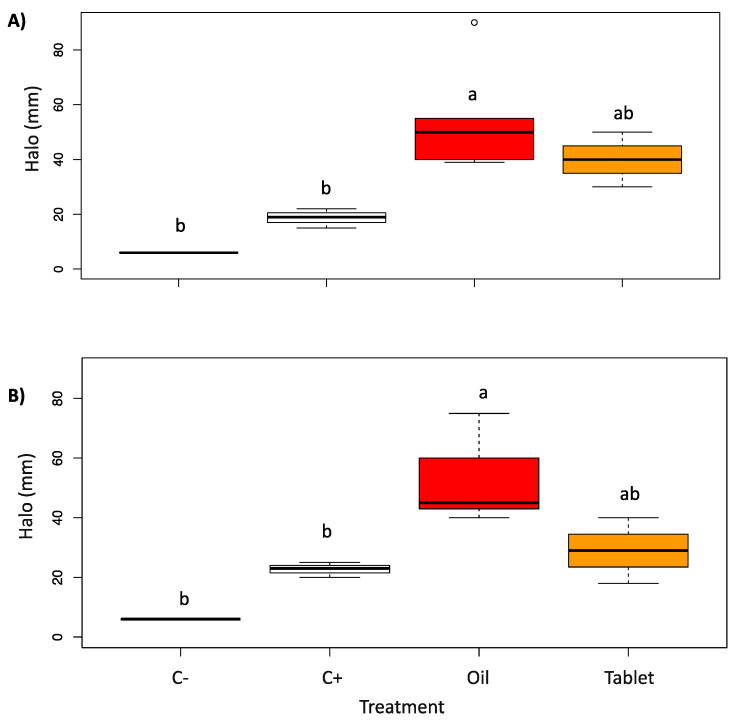
Boxplots of the growth inhibition halo (mm) induced by *T. vulgaris* on *C. albicans* (**A**) and *C. glabrata* (**B**). C- = negative control (1,4-dioxane); C+ = positive control (clotrimazole); Oil = *T. vulgaris* essential oil; Tablet was produced adding *T. vulgaris* essential oil with cyclodextrin-powder. Different letters in the figure indicate significant differences between treatments at *p* < 0.05.

**Figure 6 pharmaceutics-15-00914-f006:**
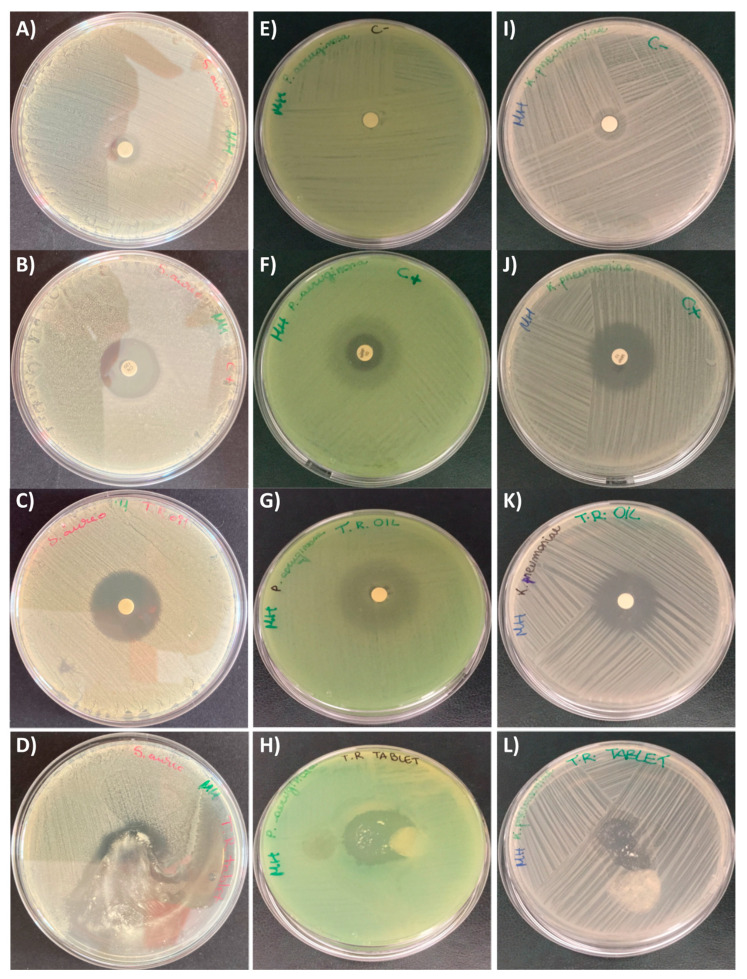
Growth inhibition halo (disk diffusion test) induced by *T. vulgaris* on *S. aureus*, *P. aeruginosa,* and *K. pneumoniae*. (**A**) *S. aureus* negative control (1,4-dioxane); (**B**) *S. aureus* positive control (gentamycin); (**C**) *T. vulgaris* essential oil against *S. aureus*; (**D**) Tablet against *S. aureus*; (**E**) *P. aeruginosa* negative control (1,4-dioxane); (**F**) *P. aeruginosa* positive control (meropenem); (**G**) *T. vulgaris* essential oil against *P. aeruginosa*; (**H**) Tablet against *P. aeruginosa;* (**I**) *K. pneumoniae* negative control (1,4-dioxane); (**J**) *K. pneumoniae* positive control (meropenem); (**K**) *T. vulgaris* essential oil against *K. pneumoniae*; (**L**) Tablet against *K. pneumoniae*.

**Figure 7 pharmaceutics-15-00914-f007:**
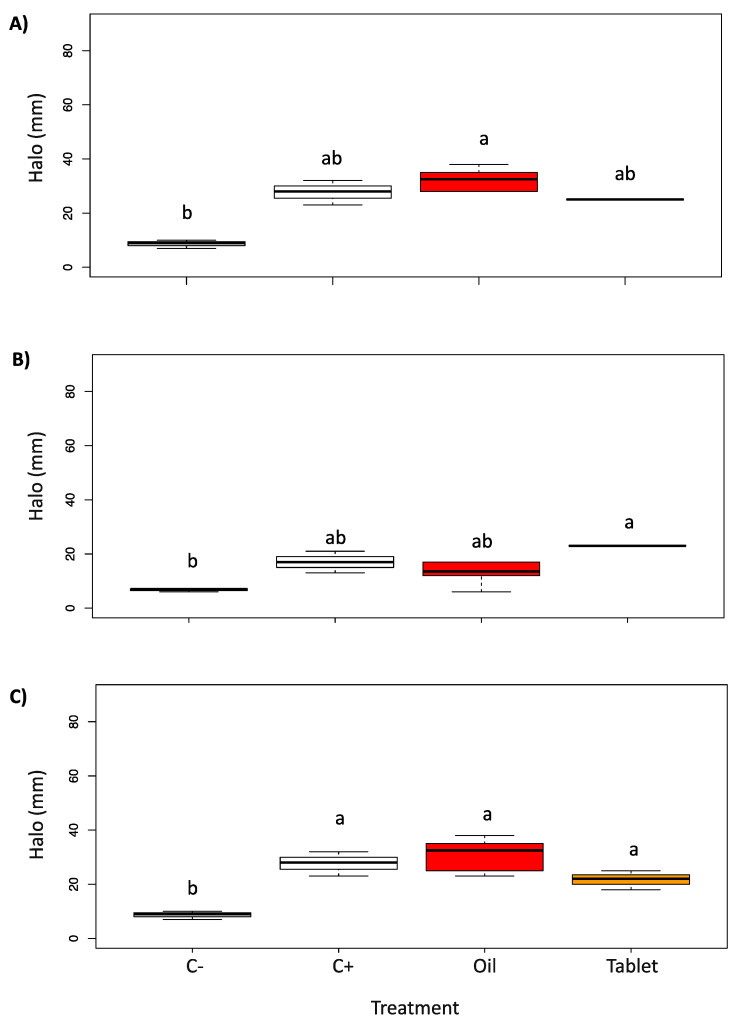
Boxplots of the growth inhibition halo (mm) induced by *T. vulgaris* on *S. aureus* (**A**), *P. aeruginosa* (**B**), and *K. pneumoniae* (**C**). C− = negative control (1,4-dioxane); C+ = positive control (gentamycin, meropenem, meropenem, respectively); Oil = *T. vulgaris* essential oil; Tablet was produced adding *Thymus vulgaris* essential oil with cyclodextrin-powder. Different letters in the figure indicate significant differences between treatments at p < 0.05.

**Table 2 pharmaceutics-15-00914-t002:** Compounds identified by GC-MS method.

CAS n°	RT [a]	Compound	Chemical Class	EO	EO + bCD	Info
17699-16-0	7.29	(E)-Sabinene hydrate	Monoterpene	X	-	-
138-86-3	7.85	Limonene	Monoterpene	X	-	Antimicrobial activities [30]. Anti-proliferative activities [31]. Antioxidant and anti-inflammatory effects [32].
470-82-6	8.07	Eucalyptol	Monoterpenoid	X	X	Anti-inflammatory, antioxidant activities [33].
99-87-6	10.13	p-Cymene	Monoterpene	X	X	Antimicrobial, anticancer, antioxidant, anti-inflammatory, antinociceptive, and anxiolytic properties [34,35].
586-62-9	10.54	Terpinolene	Monoterpene	X	X	Sedative activity [36].
na	17.10	Epoxyterpinolene	Monoterpene	X	-	
78-70-6	20.63	Linalool	Monoterpenoid alcohol	X	X	Anti-tumor, anti-cardiotoxicity activity [37].
586-82-3	21.40	α-Terpineol	Monoterpenoid alcohol	X	-	Antioxidant, antiinflammatory, anticonvulsant, antimicrobial, anticarcinogenic properties [38].
87-44-5	21.81	Caryophyllene	Bicyclic sesquiterpene	X	X	CB2 receptor agonist [39] and shows anti-cancer, antioxidant, and antimicrobial properties [40].
10198-23-9	21.96	β-Terpinyl acetate	Monoterpenoid	X		
138-87-4	23.23	β-Terpineol	Monoterpenoid alcohol	X	X	Plant metabolite, volatile oil component, and fragrance [41].
124-76-5	24.22	Isoborneol	Bicyclic monoterpenoid alcohol	X	X	Antioxidant and antiviral properties [42,43].
673-84-7	24.62	Allo-Ocimene	Monoterpene	X	X	Activate defense genes and induceresistance against Botrytis cinerea in Arabidopsis thaliana [44].
4584-65-0	24.70	5-Methyltropolone	Cyclic ketone	X	X	
80-26-2	25.03	α-Terpinyl acetate	Monoterpenoid ester	X	X	Potential antioxidant and anti-amyloidogenic activities [45].
10482-56-1	25.15	(−)-α-Terpineol	Monoterpenoid alcohol	X	X	Aroma compound [46].
586-81-2	25.26	γ-Terpineol	Monoterpenoid alcohol	X	X	
527-60-6	25.81	Mesitol	Aromatic alcohol	X	-	Probe compound shown to react mainly with organic matter (3DOM) [47,48].
3304-28-7	26.05	5-Methyl-2-(1-methylethylidene)-4-hexenal	Aldehyde	X	-	
523-47-7	26.64	β-Cadinene	Cyclotherpene	X	-	
106-22-9	27.10	(R)-(+)-Citronellol	Monoterpene	X	X	Anti-cancer activity [49].
106-25-2	27.88	Nerol	Terpene alcohol	X	X	Triggers mitochondrial dysfunction and induces apoptosis via elevation of Ca^2+^ and ROS. Antifungal activity [50,51].
55282-11-6	28.59	11-(1-Ethylpropyl)heneicosane	Alkane	0	X	
106-24-1	28.95	cis-Geraniol	Monoterpenoid alcohol	X	X	Anti-tumor, anti-inflammatory, antioxidative, and antimicrobial activities, and hepatoprotective, cardioprotective, and neuroprotective effects [52].
6994-90-7	29.90	(R-1,T-4)-4,8-Epoxy-p-menthan-1-ol	Alcohol	X	X	
na	30.46	9-Oxabicyclo[3.3.1]non-6-en-3-ylmethanol	Alcohol	X	-	
1139-30-6	30.96	Caryophyllene oxide	Bicyclic sesquiterpene	X	X	Analgesic and anti-inflammatory activity [53].
55090-55-6	31.31	Camphene hydrate-9-D	Bicyclic monoterpene	X	-	
na	31.48	Diepicedrene-1-oxide	Epoxide	X	-	
122-03-2	31.59	Cuminaldehyde	Aromatic aldehyde	X	-	A natural aldehyde with inhibitory effects on alpha-synuclein fibrillation and cytotoxicity. Cuminaldehyde shows anti-cancer activity [54].
135760-25-7	31.72	Ascaridole epoxide	Epoxide	X	-	
19888-34-7	32.08	Humulene epoxide ii	Sesquiterpene epoxide	X	-	
87096-70-6	32.17	5-(1-Hydroxy-1-methylethyl)-2-methyl-2-cyclohexene-1,4-diol	Diol	X	-	
23665-67-0	32.30	(2Z)-6,6-Dimethoxy-3-methyl-2-hexenyl acetate	Alkene	X	X	
544-76-3	33.32	Hexadecane	Alkane	X	X	
1940-19-8	32.93	1-Vinylcyclohexanol	Tertiary allylic alcohol	X	-	
na	34.06	3-Methyl-6-hydroxybenzo[C]-dihydrofuran	Isocoumarans	X	-	
89-83-8	34.28	Thymol	Monoterpene	X	X	Antioxidant, anti-inflammatory, antibacterial, and antifungal effects [55].
768-91-2	34.48	1-Methyladamantane	Polycyclic alkane	X	-	
499-75-2	34.62	Carvacrol	Monoterpenoid	X	X	Antioxidant, anti-inflammatory, and anti-cancer properties [56].
5875-45-6	35.80	2,5-di-tert-butyl-phenol	Phenol	X	X	Antioxidant [57].
55044-09-2	36.07	1-Ethyl-3-(2-[2-(3-ethylphenyl)ethoxy]ethyl)benzene	Substituted benzene	X	-	
646-31-1	36.92	Tetracosane	Straight-chain alkane	-	X	
629-99-2	37.98	Pentacosane	Straight-chain alkane	-	X	Anti-cancer activities [58].
na	38.04	2,5-Dimethylbicyclo[3.3.0]oct-6-en-8-one	Ketone	X	-	
1928-30-9	38.52	2-Methyltricosane	Straight-chain alkane	-	X	
na	38.68	6-Ethyl-5-hydroxy-2,3,7-trimethoxynaphthoquinone	Naphthoquinone	-	X	
630-01-3	38.99	Hexacosane	Straight-chain alkane	-	X	
59906-94-4	39.09	1-Methoxy-2-mesitylacenapthylene	Polycyclic aromatic ether	X	X	
71697-85-3	39.17	5-(1-Bromo-1-methylethyl)-2-methyl-2-cyclohexen-1-one	Ketone	X	X	
105314-84-9	39.26	3,9-Dimethoxy-11A-methylpterocarpan	Isoflavonoids derivative	X	-	
na	39.40	1,4-Di(tert-butylethynyl)benzene	Substituted benzene	X	-	
54725-16-5	39.56	7a-Methyl-1,4,5,6,7,7a-hexahydro-2H-inden-2-one	Ketone	X	-	
82849-65-8	39.62	5,6-C(13)(2)-1,5,9-Decatriyne	Decatriyne	-	X	
544-63-8	39.83	Myristic acid	Saturated long-chain fatty acid	-	X	
630-04-6	39.96	Hentriacontane	Long chain alkane	-	X	
93796-74-8	40.20	Ascomatic acid	Dibenzofuran	X	-	
1166-72-9	40.62	9-Thiocyanato-androst-4-en-11-ol-3,17-dione	Ketone	X	-	
na	40.80	2-Hydroperoxy-2-(2-oxiranyl)-adamantane	Hydroperoxide	X	-	
630-06-8	40.90	Hexatriacontane	Long chain alkane	-	X	
na	40.93	1-Oxa-2-oxo-3,8-dihydroxy-6-methyl-acenaphthylo[4,5-B](1-oxa-4,45-trimethyl-cyclopentane)	Naphthofuran	X	-	
na	41.20	2-(3-Acetoxy-4,4,14-trimethylandrost-8-en-17-yl)-propanoic acid	Steroid hormone derivative	-	X	Phytochemical compound [59].
502-52-3	41.26	1,3-Dipalmitoyl glycerol	Glycerol	-	X	
74199-04-5	41.53	4,5,6-Trimethoxy-3′,4′-methylenedioxybiphenyl-2-carbaldehyde	Carbaldehyde	X	-	
57-10-3	41.79	Hexadecanoic acid	Long-chain saturated fatty acid	-	X	Anti-cancer activity [60].
66205-02-5	42.70	1-n-Hexyl-7-n-butyl-1,2,3,4-tetrahydronaphthalene	Naphthalene derivative	X	-	
124821-10-9	42.75	(+−)-cis-3,4,6,9-Tetrahydro-7,10-dimethoxy-1,3,8-trimethyl-1H-naphtho [2,3-C]pyran-6,9-dione[(+−)-ventilagone-7,10-dimethyl ether]	Isochromanequinone	X	-	
80893-74-9	42.88	2-Methoxy-6-(3′,5′-dimethoxyphenyl)methylbenzoic acid	Aromatic compound	X	-	
7683-64-9	43.35	Squalene	Triterpene	-	X	Antioxidant, potential anti-cancer activities [61].
57-11-4	43.99	Stearic acid	Long-chain fatty acid	-	X	Reduction of visceral adipose tissue in athymic nude mice [62].
107971-21-1	44.96	11-Methylbenzo[3,4]phenanthro[1,2-B]thiophene	Thiophene derivative	X	-	
71013-35-9	45.64	1,8-Dimethoxy-3-methyl-anthraquinone	Anthraquinone	X	X	
33585-88-5	46.96	5,19-Cyclo-5β-androst-6-ene-3,17-dione	Steroid hormone derivative	X	-	
1166-72-9	47.10	9-Thiocyanato-androst-4-en-11-ol-3,17-dione	Steroid hormone derivative	X	-	
na	47.29	6,7-Dimethoxy-4H-cyclopenta[DEF]chrysene	Polycyclic aromatic hydrocarbon derivative	X	X	
302-79-4	47.82	Retinoic acid	Retinoid	X	X	Metabolite of vitamin A. Plays important roles in cell growth, differentiation, and organogenesis. Natural agonist of RAR nuclear receptors [63,64,65,66].
105314-88-3	48.51	2,9-Dimethoxy-4B,9B-dihydro-4B,9B-dimethylbenzofuro[3.2-B]benzofuran	Aromatic heterocycle	X	-	

[a] GC retention time.

**Table 3 pharmaceutics-15-00914-t003:** Quantitative GC analysis.

Compound	RT ^[a]^ (min)	Calibration Curve	R^2^ Value	EO (mg/g)	EO-bCD (mg/g)	Incorporation (%)	Active Molecules in Tablet (mg/Tablet)
Eucalyptol	8.06	y = 66.402x − 38.233	0.9999	57.1	0.022	1.7	0.44
Linalool	20.62	y = 72.665x − 47.210	0.9996	71.4	0.018	3.9	0.36
trans-Caryophyllene	21.80	y = 96.497x − 64.083	0.9994	60.0	0.011	7.0	0.22
Thymol	34.26	y = 96.760x − 60.319	0.9995	60.0	0.161	6.4	3.22
Carvacrol	34.62	y = 115.694x − 74.025	0.9994	80.0	0.042	6.6	0.84

^[a]^ = GC retention time.

## Data Availability

Not applicable.

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
