# Peer review of "Thymus vulgaris Essential Oil in Beta-Cyclodextrin for Solid-State Pharmaceutical Applications"

_pharmaceutics, 2023, doi:10.3390/pharmaceutics15030914_

Round 1

Reviewer 1 Report

The paper pharmaceutics-2216670 describe the use of cyclodextrin entrapped thymus vulgaris essential oil for improved performance solid state pharmaceutics against fungi and bacteria. The paper is well written and may be published after revision. The strength of the paper consists in the finding that CD trapped essential oil tablets, obtained by simple green technology are effective against pathogens.

Some suggestions for the authors to improve the work:

1.       At the end of the abstract emphasize originality.

2.       Give TR EO abbreviation at line 219.  Define RT in table 2. Define/correct tr in the table 3.

3.       Check English and typos “table34” at line 296 should be” table 3”

4.       Spectral resolution is not the same with accuracy of transmittance/wavenumber values. Please see line 234. Instead of referring to resolution give the accuracy if available. Spectral shifts as low as 0.04 cm-1 or less depending on the instrument may be considered.

5.       Indicate FTIR peaks in the figures 1 and 2 (at least the peaks discussed in the text),

6.       The C-H aromatic stretching bands are hardly observed in the b-CD TR EO complex. Check line 247. They are almost covered by the wide OH stretching band of CD.

7.       Try to infer from the FTIR data, spectral shifts, the possible interaction modes of the EO and BCD

8.       Give OE abbreviation at line 110.

9.       Correct Figure 3 caption. Increase axis text size, it is illegible. Give the three tg’s superposed in one image for easy comparison.

10.   Compare the tg results with data in the literature.

11.   How is incorporation calculated in table 3?

Author Response

Dear reviewer,

Thank you very much for your comments. We have followed the suggestions and advice while effectively improving the manuscript.

Comments and Suggestions for Authors

The paper pharmaceutics-2216670 describe the use of cyclodextrin entrapped thymus vulgaris essential oil for improved performance solid state pharmaceutics against fungi and bacteria. The paper is well written and may be published after revision. The strength of the paper consists in the finding that CD trapped essential oil tablets, obtained by simple green technology are effective against pathogens.

Some suggestions for the authors to improve the work:

  1. At the end of the abstract emphasize originality.

Answer: Done.

  1. Give TR EO abbreviation at line 219.  Define RT in table 2. Define/correct tr in the table 3.

Answer: The abbreviations were corrected and defined.

  1. Check English and typos “table34” at line 296 should be” table 3”

Answer: Done.

  1. Spectral resolution is not the same with accuracy of transmittance/wavenumber values. Please see line 234. Instead of referring to resolution give the accuracy if available. Spectral shifts as low as 0.04 cm-1 or less depending on the instrument may be considered.

Answer: The spectral resolution was chosen to be 2cm-1 since it is a good compromise between background noise, measurement time and accuracy of the observed peak shifts which are in the 1-15 cm-1 range.

  1. Indicate FTIR peaks in the figures 1 and 2 (at least the peaks discussed in the text),

Answer: Done.

  1. The C-H aromatic stretching bands are hardly observed in the b-CD TR EO complex. Check line 247. They are almost covered by the wide OH stretching band of CD.

Answer: A discussion about this topic is added in the manuscript

  1. Try to infer from the FTIR data, spectral shifts, the possible interaction modes of the EO and BCD

Answer: A discussion about this topic is added in the manuscript

  1. Give OE abbreviation at line 110.

Answer: The letters were switched to EO

  1. Correct Figure 3 caption. Increase axis text size, it is illegible. Give the three tg’s superposed in one image for easy comparison.

Answer: Done.

  1. Compare the tg results with data in the literature.

Answer: Done

  1. How is incorporation calculated in table 3?

Answer: a method description was added in the manuscript

Reviewer 2 Report

This study reports the results of Thymus vulgaris Essential Oil in Beta-Cyclodextrin for Solid-State Pharmaceutical Applications”. The manuscript is well prepared and the subject is interesting. In my opinion, the manuscript is in a position to be accepted for publication after minor revision. Here are some comments on the manuscript:

General remarks:

-          Line 173 : Sabouraud agar is the optimal medium for Candida strains not “Mueller-Hinton agar”

-          Line 179: 28 °C is the optimal temperature of incubation for Candida strains instead of 37°C

-          Authors should discuss the chemical characterization of their samples with previous works.

Author Response

Reviewer 2

Dear reviewer,

Thank you very much for your comments. We have followed the suggestions and advice while effectively improving the manuscript.

This study reports the results of “Thymus vulgaris Essential Oil in Beta-Cyclodextrin for Solid-State Pharmaceutical Applications”. The manuscript is well prepared and the subject is interesting. In my opinion, the manuscript is in a position to be accepted for publication after minor revision. Here are some comments on the manuscript:

General remarks:

-          Line 173: Sabouraud agar is the optimal medium for Candida strains not “Mueller-Hinton agar”.

Answer: Standard M44-A method suggested the Mueller-Hinton Agar added with 2% Glucose and 0.5 μg/mL Methylene Blue Dye (GMB) as reference medium. So the authors used this medium.

-          Line 179: 28 °C is the optimal temperature of incubation for Candida strains instead of 37°C.

Answer: All clinical microbiologists who normally isolate Candida species use 37 °C as the incubation temperature. In fact, the reference method quoted proposes 37°C or better 35°C+- 2°C as the incubation temperature.

-          Authors should discuss the chemical characterization of their samples with previous works.

Answer: Done.